# BICOS—An Algorithm for Fast Real-Time Correspondence Search for Statistical Pattern Projection-Based Active Stereo Sensors [†]

**Patrick Dietrich [1,2,\*], Stefan Heist [1,2], Martin Landmann [1,2], Peter Kühmstedt [1] and Gunther Notni [1,3]**

[1] Fraunhofer Institute for Applied Optics and Precision Engineering IOF, Albert-Einstein-Str. 7, 07745 Jena, Germany

[2] Institute of Applied Physics, Abbe Center of Photonics, Friedrich Schiller University Jena, Albert-Einstein-Str. 15, 07745 Jena, Germany

[3] Group for Quality Assurance and Industrial Image Processing, Department of Mechanical Engineering, Ilmenau University of Technology, Gustav-Kirchhoff-Platz 2, 98693 Ilmenau, Germany

[\*] Correspondence: patrick.dietrich@iof.fraunhofer.de

[†] This work is an extended version of our research published in 2019 at the conference "SPIE Dimensional Optical Metrology and Inspection for Practical Applications VIII" held in Baltimore, MD, USA, 16–17 April 2019.

**Abstract:** Pattern projection-based 3D measurement systems are widely used for contactless, non-destructive optical 3D shape measurements. In addition, many robot-operated automation tasks require real-time reconstruction of accurate 3D data. In previous works, we have demonstrated 3D scanning based on statistical pattern projection-aided stereo matching between two cameras. One major advantage of this technology is that the actually projected patterns do not have to be known a priori in the reconstruction software. This allows much simpler projector designs and enables high-speed projection. However, to find corresponding pixels between cameras, it is necessary to search the best match amongst all pixels within the geometrically possible image area (that is, within a range on the corresponding epipolar line). The well-established method for this search is to compare each candidate pixel by temporal normalized cross correlation of the brightness value sequences of both pixels. This is computationally expensive and interdicts fast real-time applications on inexpensive computer hardware. We show two variants of our algorithm "Binary Correspondence Search" (BICOS), which solve this task in significantly reduced calculation time. In practice, our algorithm is much faster than traditional, purely cross-correlation-based search while maintaining a similar level of accuracy.

**Keywords:** real-time; active stereo vision; 3D measurement; GOBO projection; statistical pattern projection; aperiodic sinusoidal fringes; GPGPU; BICOS; correspondence search; binary features

## 1. Introduction

### 1.1. Stereo Vision Based 3D Sensors

Stereo vision based 3D measurement setups are being applied in the industry for many years. Applications include, for example, reverse engineering, digitization of cultural heritage objects [1], medicine [2]. These 3D sensors triangulate 3D points from pixel correspondences between two cameras or between a camera and a projector. They can be classified into passive systems which work on images taken with ambient illumination only, and active systems with an illumination unit which projects patterns onto the measurement object.

Active illumination stereo vision systems can be classified in three different ways:

1. **Single-shot vs. multi-shot.** Single-shot systems work on a single image (or image pair for stereo camera systems) and a fixed projection pattern. Multi-shot systems record a sequence of images (or sequence of image pairs). The projected pattern is different for each successively taken image/image pair.
2. **Single-camera vs. multi-camera.** Systems which use a single camera find correspondences between the camera and the projector. Multi-camera systems use at least two cameras and find correspondences between them.
3. **Coded light vs. statistical patterns.** Coded light systems project well-known patterns or pattern sequences onto the measurement object. An overview of coded light techniques can be found in Reference [3]. On the other hand, statistical pattern systems do not require well-known patterns but can work with quasi random patterns which do not need to be known to the reconstruction algorithm.

The Microsoft Kinect sensor is an example of a single-shot, single-camera, coded light system [4]. Classical photogrammetry could be interpreted as a single-shot, multi-camera, statistical pattern system which utilizes the measurement object's texture as a pattern.

One of the most popular 3D sensor concepts is phase-shift profilometry, also called digital fringe projection with phase-shifting or phase-shift interferometry [5]. It is a multi-shot, coded light technique [6], typically implemented with a single camera but sometimes extended with additional cameras [7]. The projection patterns are (periodic) sinusoidal fringes which are shifted by a fixed phase-offset with each successive projection [6]. Some systems project additional patterns for fringe-phase unwrapping [8]. Algorithms for phase-shift pattern processing are a field of high research activity [9–11]. A correspondence between camera and projector can be found by decoding the brightness values of a single camera pixel. The 3D reconstruction only works on a very specific pattern sequence. In other words, information about the actually projected pattern sequence is required as an input for the reconstruction algorithm. Therefore, phase-shift profilometry (and other coded light methods) need a very well controllable and calibrated projector [12].

In contrast, the statistical pattern method makes this requirement unnecessary. It is always implemented with multiple cameras (usually two). It can be implemented as a single-shot system [13] or as a multi-shot system which yields higher accuracies [14]. The main reason for using statistical patterns (instead of coded light) is the possibility to build simpler projectors. The projector requirements are lower compared to the coded light method: the patterns just need to have "sufficient" spatial and temporal variation but the system does not rely on a known pattern sequence. Projection methods include, for example, laser speckle projection [13,15], projection of band limited random patterns [15,16], and projection of aperiodic sinusoidal fringes [17,18].

In the past years, we have developed several multi-shot, dual camera, statistical pattern sensor systems. This technique has proved to work well, especially for applications where commercial projectors are not available (e.g., extended spectral ranges, high camera frame rates of 10 kHz or higher). We have realized irritation free (near infrared—NIR) facial measurement systems [19], high speed measurement systems to measure fast human body motions [20] or air-bag inflations [21], and a system with a thermal laser projector to measure glass and transparent plastic objects [22].

However, these systems also have disadvantages. While in phase shift profilometry, a pixel correspondence can be found by decoding the brightness values of a single pixel, with statistical patterns, pixel correspondences between the cameras must be found by searching the best match amongst a set of candidates. This is computationally more expensive.

Some applications require that a 3D result is available within a short time after image acquisition, that is, these applications have real-time requirements. Examples include continuous position monitoring of patients during therapy, face measurement at security checks or 100%–inline quality control during industrial production. The latency between image acquisition and the availability of the

resulting 3D model must be low. Therefore, the reconstruction of the 3D model from the camera images must be fast. To satisfy this requirement, we present a real-time correspondence search algorithm for multi-shot, dual-camera, statistical pattern sensor systems in this paper.

### 1.2. Existing Algorithms and Algorithms in Related Fields

For passive stereo vision, a large research field of correspondence search algorithms exists [23], including many algorithms targeting real-time applications. However, the conditions for our active multi-shot systems are different (Table 1). We do not share many of the challenges which exist for the passive case.

**Table 1.** Data processing challenges: differences between passive stereo and active multi-shot stereo with statistical patterns.

| Passive Stereo | Active Multi-Shot Stereo with Statistical Patterns |
|---|---|
| All information for correspondence search must be found from spatial features, for example, texture, object edges, shadows, etc. | We can rely on temporal features. For a given pixel in the left camera we can find a correspondence without looking at any other pixel in the same camera. |
| Correspondences for smooth image areas without texture or other features must be guessed from surrounding image features. | The projected patterns are visible on all object parts, pixel-wise correspondences can also be found in smooth image areas. |
| Spatial image features look slightly different from each camera perspective, they have a different projected geometry. The correspondence search algorithm must account for that. | If only temporal features are used (i.e., only the brightness value sequence of a single pixel), there is only minimal geometric change. |
| The reflected ambient light intensity may be different at each camera view-point because the reflectivity of most objects depends on the angles towards light source and view point. | In addition to (unwanted) ambient light, the projected patterns may have a different intensity for each camera. The reflection factors for the projected light and the ambient light are different, because the angles towards the light sources are different. |

To find correspondences, a similarity measure is required. In the passive case, the similarity measure compares a region of pixels in the first camera with a region in the second camera [23]. In the active case, this is an option too, but more often the grey value sequences of a single pixel from each camera are compared (i.e., purely temporal) which yields higher accuracies [14]. A combination of both (a small spatial window for each image in the temporal sequence) has also been applied successfully [15].

Similarity measures for passive stereo include, for example, the Sum of Absolute Differences (SAD), the Sum of Square Differences (SSD) and the Normalized Cross Correlation (NCC) [23–25]. For the active case the NCC is by far the most popular similarity measure [14–16]. Other measures have been tried: SAD, SSD, correlation of temporal gradients, Sum of Absolute Differences of temporal gradients; but NCC yields the most accurate results [26].

In this paper, we propose to split the correspondence search into two parts: 1. a coarse correspondence search which is based on our newly developed algorithm and 2. a correspondence refinement step which uses NCC. The result from the first step is used as a initial solution for the refinement step. In this way, we can significantly improve the speed of the correspondence search without compromising the measurement accuracy.

We have proposed one variant of our new algorithm "Binary Correspondence Search (BICOS)" in Reference [27]. Since then, we have improved it (BICOS+) and now present an evaluation of both algorithm variants. We compare it against a reference algorithm which is purely NCC based.

Our algorithm uses binary features (BF) to describe the temporal brightness value sequence of a pixel. Binary features have also been proposed for correspondence search in photogrammetry between uncalibrated cameras [28–30] and have been used extensively for texture classification [31].

In these applications, binary features are used to describe a spatial image area whereas we use them to describe a single pixel in a temporal image stack.

## 2. Materials and Methods

### 2.1. General Reconstruction Algorithm Outline

The reference algorithm (see Section 2.2) as well as our new algorithm (BICOS and BICOS+) have the following outline (Figure 1).

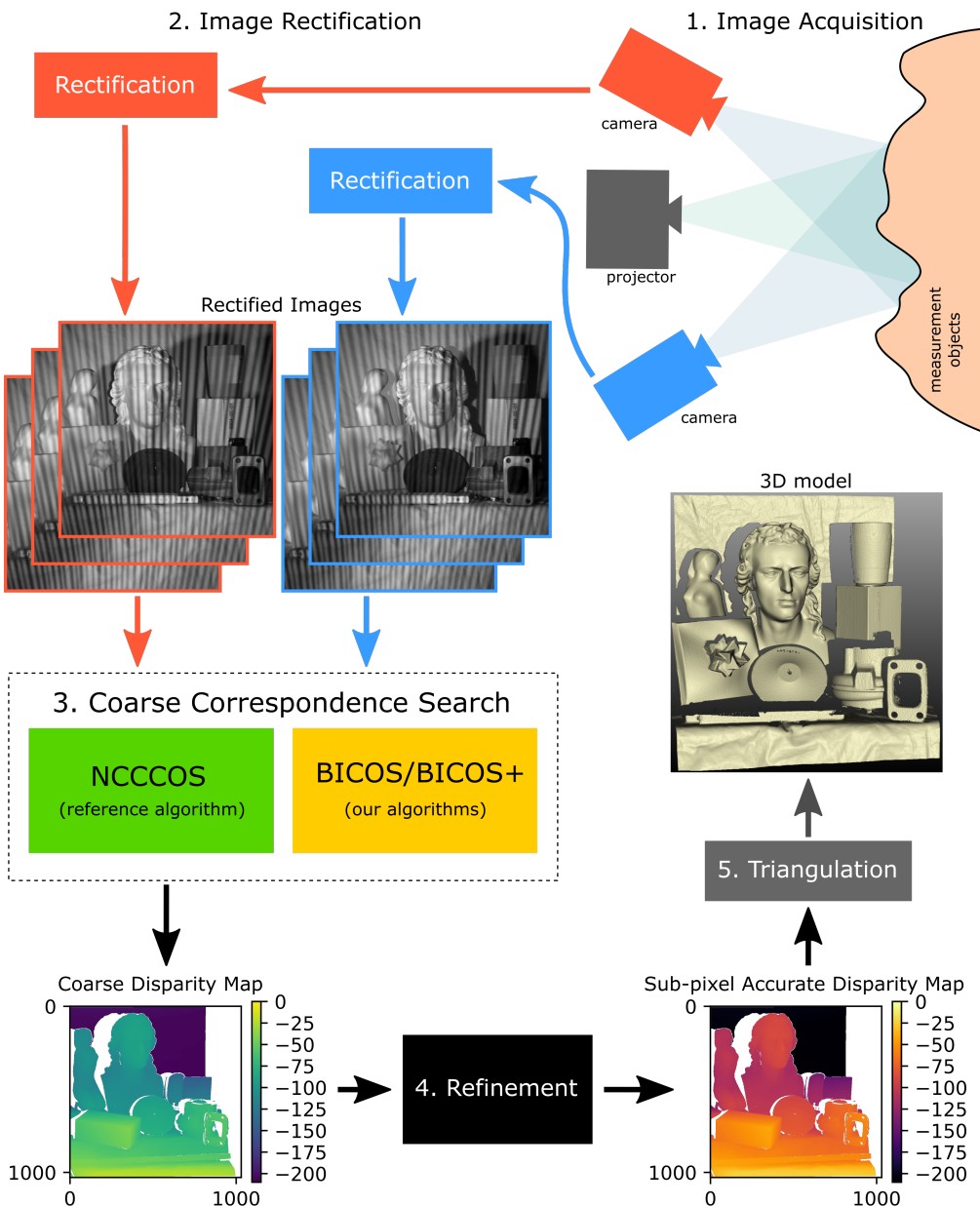

**Figure 1.** General outline of the 3D reconstruction.

1.  **Image Aquisition.** A sequence of image pairs is recorded synchronously with the two cameras of the stereo setup.
2.  **Rectification.** We rectify the camera images, that is, we apply a geometric transformation to the images to simulate two cameras with parallel image planes. The rectification algorithm also corrects lens distortion. After rectification, the images have the important property that an object point, which is visible in a specific image row $r$ of the first camera's images, also appears in the

identical row *r* in the other camera's images. This means that the epipolar lines [32,33] are parallel to the image rows.

3.  **Coarse Correspondence Search.** We search corresponding pixels between the cameras. For each pixel in the left camera at position $(c_L, r)$, we search a pixel in the right camera at position $(c_R, r)$ which shows the same object point. Due to rectification (step 2), this pixel can be found on the same image row *r*. The column $c_R$ has to be determined. In the coarse correspondence search, we consider any $c_R$ with a distance of up to two pixels from the real correspondence correct. The difference of the column numbers $d = c_L - c_R$ is called "disparity". The result of the coarse correspondence search is the coarse disparity map.

4.  **Correspondence Refinement.** We refine the result by searching an interval around the coarse correspondence. We then interpolate between pixels in the right image and find the best match amongst the interpolated sub-pixels. The result is the refined disparity map.

5.  **Calculation of 3D points.** For each pixel in the disparity map, we calculate a 3D point by triangulation.

Open-source code which facilitates writing algorithms based on this approach is readily available [34,35]. As we work with statistical patterns, we do not have a-priori information about the projected patterns. This means that, in the coarse correspondence search (step 3), the best matching pixel needs to be searched amongst all candidates. In practice, this is the most time consuming part of the algorithm. We therefore focus on the coarse correspondence search in this paper. Its goal is to find the approximate position of the correspondence up to an accuracy of a few pixels. All further correspondence refinement (step 4) happens afterward in a separate algorithmic step where sub-pixel accuracy is achieved.

Starting with a pixel in the left camera's rectified image sequence, we search a set of candidate pixels in the same row of the right camera's rectified image sequence. The set of candidate pixels is determined by the sensor geometry and the size of the measurement volume. This means that only a restricted interval of disparities have to be searched.

To compare two pixels, a measure of similarity is required. This is where the algorithms differ from each other (Sections 2.2 and 2.3). The pixel which is the most similar according to the measure is picked amongst the candidates.

When the matching pixel has been found (in the right camera's image sequence), we validate it by reverse searching the best match for it in the left camera image sequence. Only if the so found reverse match is equal (or at a maximum distance of 2) to the original pixel, the match is accepted.

*2.2. Reference Algorithm (NCCCOS)*

The reference algorithm uses normalized cross correlation (NCC) as the similarity measure [14,16]. We therefore call it NCC-based correspondence search (NCCCOS). To compare two pixels, the cross correlation between the temporal sequences of the brightness values of those pixels are calculated. The NCC takes on values from $-1.0$ to $1.0$. A cross correlation of $1.0$ means that the two pixels match perfectly; the lower the cross correlation, the less the similarity.

The coarse correspondence search is far more time-consuming than any other part of the algorithm. This has two reasons: the number of comparisons between pixels is huge and each such comparison is computationally expensive. There are two possible ways to improve computation speed for the coarse search:

1.  reduce the number of pixel comparisons (e.g., by further restricting the scan volume)
2.  reduce the calculation time which is required for each pixel comparison

We take the latter approach with our BICOS and BICOS+ algorithms, that is, the number of pixel comparisons is the same but the pixel comparisons are faster in our algorithms.

### 2.3. BICOS Algorithm

This section contains the details of our BICOS algorithm. We add a pre-calculation before the actual correspondence search which allows us to reduce the number of required calculations for each pixel comparison. Our coarse correspondence search comprises three steps:

1. **Calculation of binary features.** For each pixel, we first calculate a bit string of "binary features". A binary feature is generated by comparing two of the brightness values of a pixel with each other (Figure 2a,b). For example, comparing brightness values $b_1$ with $b_2$ yields a binary feature with a value of 1 if $b_1 > b_2$ and 0 otherwise. It does not matter, if a $>$, $<$, $\geq$ or $\leq$ operator is used, as long as it is consistent.

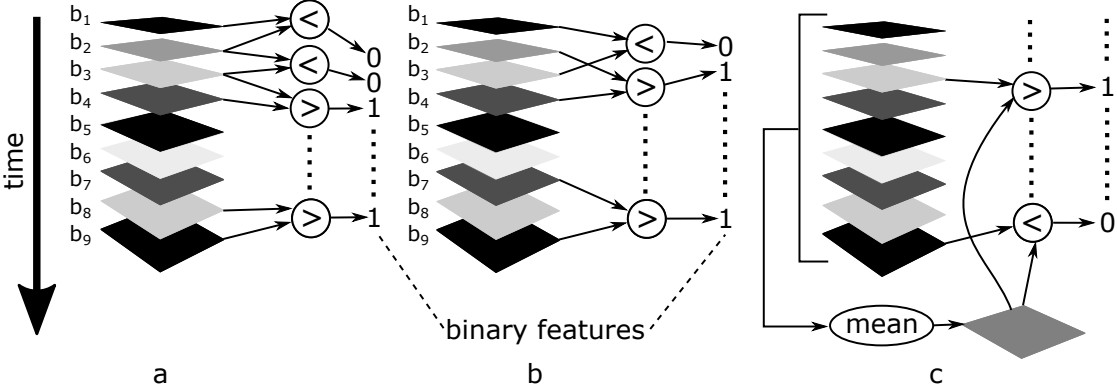

**Figure 2.** Creation of binary features from a pixel's sequence of brightness values. (**a**) Comparison of temporally adjacent brightness values. (**b**) Comparison of each brightness value with every second brightness value. (**c**) Comparison of each brightness value with the mean brightness of the pixel (BICOS and BICOS+).

We can compare every brightness value with every other brightness value within the sequence. For instance, for a sequence length of 10, we compare brightness value $b_1$ with $b_2$, $b_2$ with $b_3$, ..., $b_9$ with $b_{10}$, $b_{10}$ with $b_1$, $b_1$ with $b_3$ and so forth. This yields 45 binary features. In addition, each brightness value can be compared to the mean brightness value of the sequence, yielding another 10 binary features for a sequence length of 10 (Figure 2c). We restrict the number of binary features to 64 to allow fast computation.

2. **Comparing the binary features.** To find a correspondence we do not compare the brightness values but we compare the binary features of a pixel in the left camera with the binary features of a pixel in the right camera. The more binary features coincide, the better the match. Thus the pixel similarity measure for BICOS (and BICOS+) is the Number of Equal Binary Features (NEBF). Calculation of the NEBF is very fast because only two operations are required: compare and count. Like in the NCCCOS, we perform the correspondence search in both directions, that is, first from left to right, then from right to left. We accept only consistent results.

3. **Filtering the coarse correspondences.** The coarse disparity map which we generate by using the NEBF contains more outliers and holes than the one created with the NCC which is used in NCCCOS. For an explanation why this is the case, see Section 3.2. We compensate for this by applying a $3 \times 3$ median filter to the disparity map, which also fills the holes. Please note that this does **not** mean that the final 3D result is filtered because we apply this filter only to the coarse disparity map which is used as a start solution for the refinement step.

### 2.4. BICOS+ Algorithm

The difference between the original BICOS and the improved BICOS+ algorithm is the type of binary features which we use. Both algorithms use the binary features which are calculated by comparison of single brightness values with the mean brightness value (Figure 2c). However, the other binary features in BICOS+ are calculated with the following rule which we show exemplary in Figure 3.

1. form pairs of brightness values (from the brightness value sequence of a pixel)
2. sum up the two brightness values of each pair
3. compare two such sums with each other and save the result of the comparison as a binary feature (only sums which do not share a common brightness value are compared)

The main advantage of this method over the original BICOS is the higher number of features which can be created this way (Figure 4). Only for sequence lengths of 6 and lower this method yields less than 64 binary features. For these short sequence lengths, the missing binary features are filled up with other binary features from direct comparisons of 2 brightness values (as in the original BICOS algorithm, Figure 2a,b).

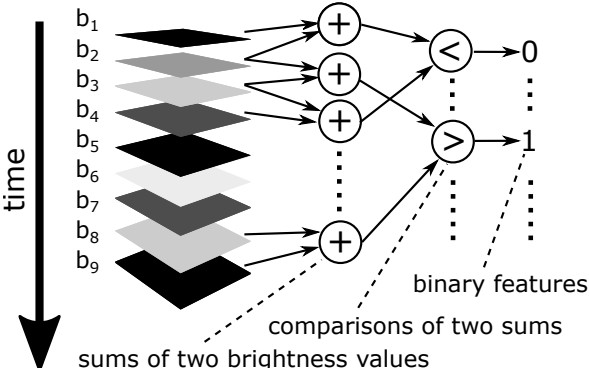

**Figure 3.** Creation of binary features from two sums of brightness values (BICOS+).

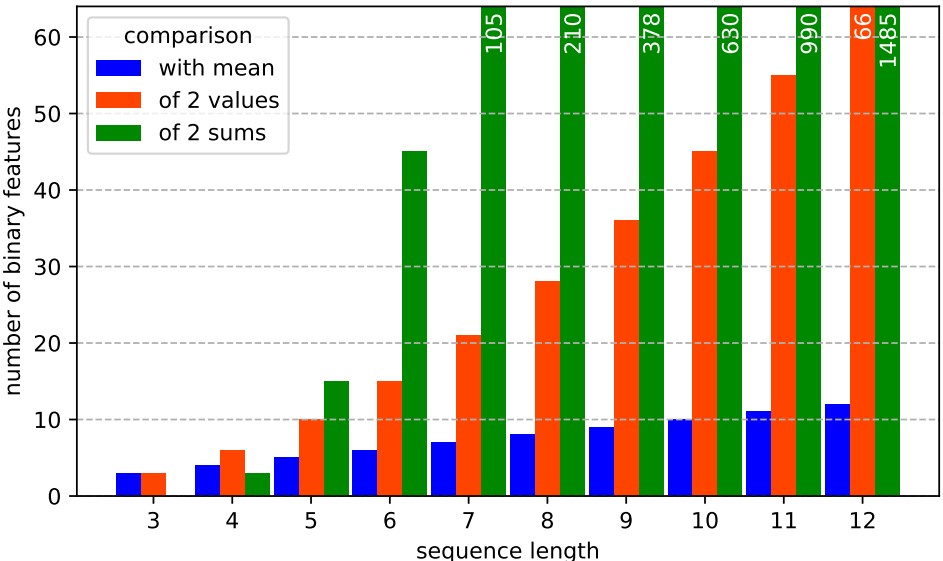

**Figure 4.** Number of binary features (BF) by type. Blue: BF created by comparing single brightness values with the mean brightness of the pixel. Orange: BF created by comparing 2 brightness values of the sequence with each other. Green: BF created by comparing two sums with each other, which have each been calculated from two brightness values (only for BICOS+).

### 2.5. Robustness of the Algorithms against Ambient Light and Changes in Reflectivity

The NCC is invariant under the following changes from left to right camera: ambient light, object reflectivity and camera sensitivity [26]. This means, it has the following property:

$$NCC(\vec{a}, \vec{b}) = NCC(\vec{a}, s \cdot \vec{b} + A) \qquad \forall s > 0$$

where $\vec{a}$ and $\vec{b}$ are the two brightness sequences in the left and right image sequence, $s$ represents the deviation in camera sensitivity and reflectivity of the measured object towards the left and the right camera, $A$ represents ambient light which may be different in the left and right camera.

A similar property can be derived for the NEBF. Each binary feature is calculated by comparing two brightness values of the same pixel. Any additional temporally invariant ambient light is added to both values and thus does not change the outcome of the comparison. In addition, each binary feature is scaling invariant. In other words,

$$b_1 < b_2 \Leftrightarrow (s \cdot b_1 + A) < (s \cdot b_2 + A) \qquad \forall s > 0$$

and therefore

$$NEBF(\vec{a}, \vec{b}) = NEBF(\vec{a}, s \cdot \vec{b} + A) \qquad \forall s > 0$$

However, both object reflectivity and ambient light have an influence on the signal to noise ratio (see Section 2.7).

### 2.6. Sensor for Image Acquisition

We use a 3D sensor which is composed of two near-infrared (NIR) cameras and a GOBO projector in between [19]. The scanner is depicted in Figure 5. The projector comprises a rotating slide wheel (GOBO) with a random binary stripe pattern, an LED illumination unit with a center frequency at 850 nm wavelength, and a projection lens. The binary stripe pattern becomes sinusoidal in the camera images due to lens blur and motion blur because the slide wheel is rotating during image integration time.

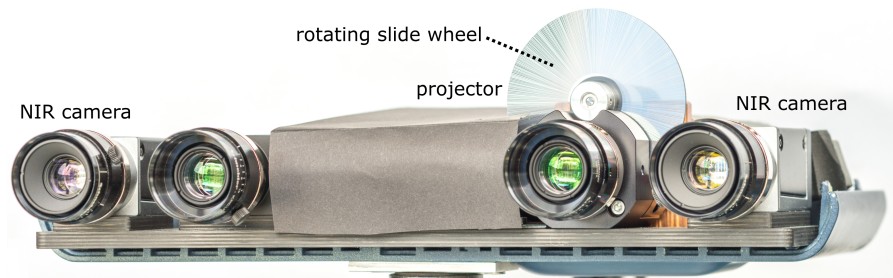

**Figure 5.** 3D sensor (with removed outer housing) which we used for data recording. From left to right: near infrared (NIR) camera, color camera (not used for this paper), NIR GOBO projector with rotating slide wheel, second NIR camera. The NIR cameras acquire the image sequences for the 3D reconstruction.

We measured the noise level of the cameras with the following method. We recorded 400 images for each camera of a static scene with stopped slide wheel. For each camera, all images contain the same scene with the same illumination except for noise. For each pixel, we calculated the temporal mean brightness value (over the 400 images) and the temporal deviation from that mean. The deviation is a measure for the noise. We show the deviation as a function of the mean in Figure 6. Please note that the measured noise levels for high brightnesses are too low because of clipping when the sensor pixels become saturated.

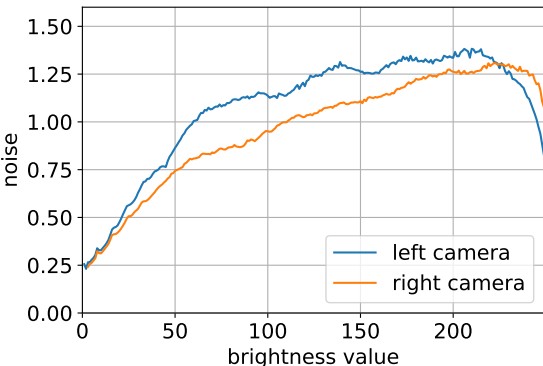

**Figure 6.** Noise levels of the rectified camera images.

### 2.7. Test Scene and Ground Truth Data

We set up a test scene with a size of about $0.4 \times 0.4 \times 0.4$ m$^3$ (Figure 7). It consists of several objects with different surface materials: Two ceramic busts with matt surface and slightly different reflectivities, a wooden box, a ceramic cup with a glossy surface and a felt handle, a 3D printed intricate plastic model, a cast iron disk and a rusty cast iron turbine housing. The scene is placed on an industrial fabric. We added an inhomogeneous NIR ambient light source from the left side of the scene to simulate real world conditions.

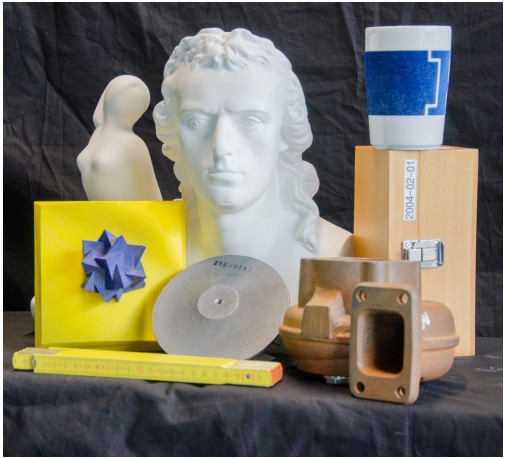

**Figure 7.** Photograph of the test scene.

We recorded a data set of 400 image pairs at a resolution of 1 Megapixel per image while the projector was running. Each image pair contains a different aperiodic sinusoidal stripe pattern (Figure 8). We rectified the images and worked exclusively with this rectified version throughout the survey. These rectified images are available for download as Supplementary material.

We calculated the temporal mean (Figure 9a) and temporal standard deviation (Figure 9b). The ambient light which is temporally constant is visible in the mean but not in the standard deviation. Because the NCC and the NEBF both work on temporal brightness contrast, the standard deviation represents the signal strength for the reconstruction algorithm. We looked up the noise level (as shown in Figure 6) for each pixel in the mean image. We then divided the temporal standard deviation by it. The result is an approximated signal to noise ratio (ASNR) (Figure 9c). The more ambient light, the higher the mean and therefore the higher the noise. This means that the ambient light has a negative influence on the ASNR.

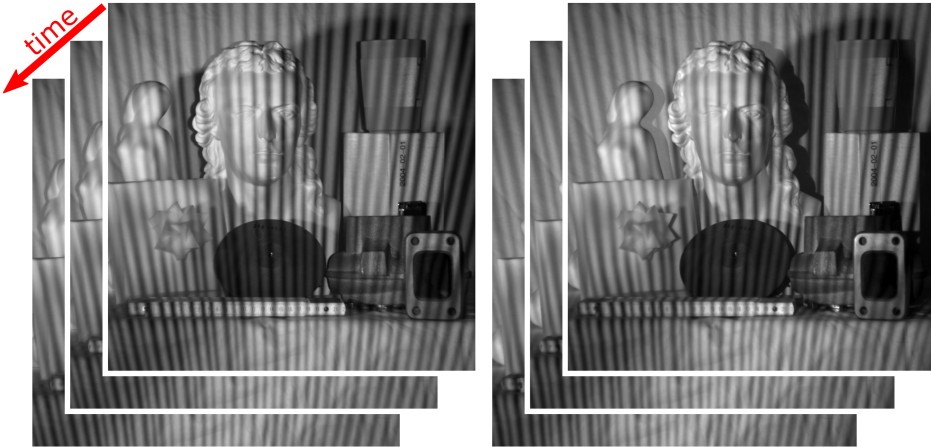

**Figure 8.** Rectified images of the left and right camera (brightness and contrast adjusted for better visibility).

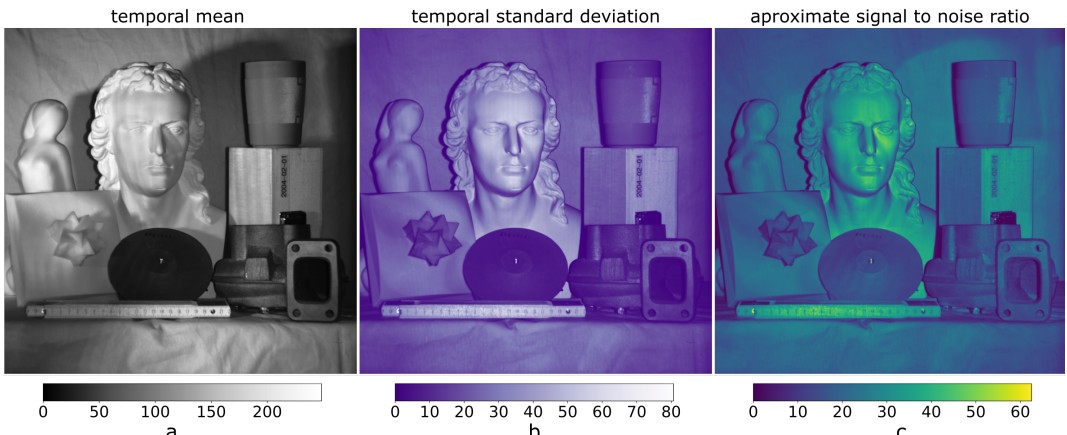

**Figure 9.** (**a**) mean of 400 images with changing projected pattern. Note the inhomogeneous ambient light coming from the left and the shadows thrown by it on the background farbic. (**b**) temporal standard deviation of the same images. Note the absence of the (temporally constant) ambient light. (**c**) approximated signal to noise ratio (ASNR). Note the negative influence of the ambient light on the ASNR value on the background fabric.

We created an accurate reference 3D model which we consider ground truth (Figure 10). We used the reference algorithm (NCCCOS) with the full sequence of 400 image pairs to calculate it.

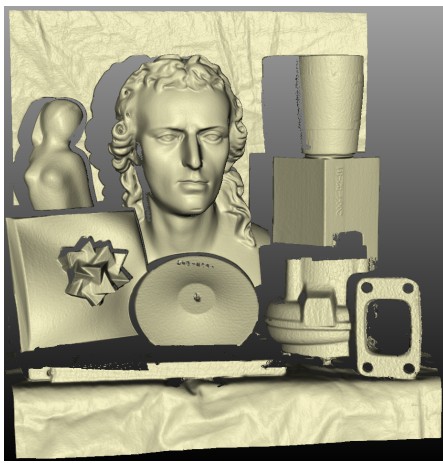

**Figure 10.** Ground truth 3D model calculated from 400 image pairs.

## 3. Results

### 3.1. Relationship between NCC and NEBF

We examined the relation between the two pixel similarity measures, the normalized cross correlation (NCC) and the number of equal binary features (NEBF), that is, what NEBF can be expected for a given NCC.

We used a sequence of 10 image pairs and the original BICOS algorithm, which has 55 binary features per pixel for this sequence length. We compared each pixel of the left camera with every candidate pixel of the right camera and stored the NCC and NEBF values for each comparison. From this data, we created a 2D histogram by dividing the data set into bins with a certain NEBF and a certain range of NCC (Figure 11).

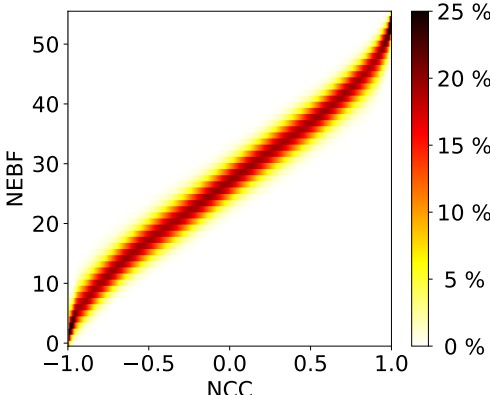

**Figure 11.** Relation between the two pixel similarity measures. 2D histogram of all pairs of normalized cross correlation (NCC) and Number of Equal Binary Features (NEBF) for all pixel comparisons of a measurement with a sequence length of 10 image pairs. For better visibility, the histogram is normed to the total number of pixels which fall in the same cross correlation bin.

The two measures show a quasi-linear relation for a large part of the value ranges. In the important range, where the two compared pixels are very similar (i.e., a NCC of close to 1.0), the "curve" becomes steeper.

### 3.2. How Well Can NCC and NEBF Distinguish between Correct and Wrong Correspondences?

We assessed how well each of the measures (NCC and NEBF) can distinguish between the correct correspondence and wrong correspondences. We calculated the differences between the measure value at the correct correspondence (according to the ground truth) and at the most similar wrong correspondence. We call this difference NCC margin and NEBF margin, respectively. For example, for a given pixel of the left camera, the NEBF might be 62 at the correctly corresponding pixel of the right camera and the best NEBF value for any pixel which is more than two pixel away from the truly corresponding pixel might be 59; the NEBF margin is then $62 - 59 = 3$. Large positive values mean that the true correspondence can be well distinguished from wrong correspondence candidates. Negative values mean that the wrong correspondence has a better measure value than the correct correspondence, which leads to a mismatch. It is also possible that several candidate correspondences have the same measure value (especially for the discrete NEBF). In that case, the margin is 0 and there are several equally good correspondence options. (The algorithm picks the first one. If this pick was wrong, it is likely to be removed in the reverse search. See Section 2.1.) We calculated the histograms of the margins for several sequence lengths (Figure 12).

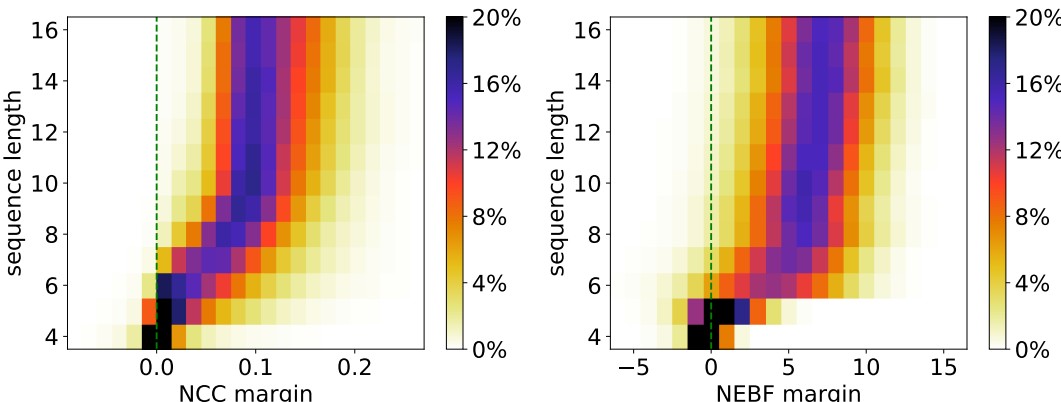

**Figure 12.** Histogram of margins of the NCC (**left**) and the NEBF (**right**) at the true correspondence vs. the most similar pixel amongst the wrong correspondences. Large positive values mean that the true correspondence can be well distinguished from wrong correspondence candidates. Negative values mean that the wrong correspondence has a better NCC/NEBF value than the correct correspondence which leads to a mismatch. For the NEBF calculation, the binary features of BICOS+ were used.

The NEBF shows more margin values close to zero than the NCC. This means that for some pixels, the NEBF performs worse at distinguishing between correct and wrong correspondences. It has more points with margins < 0. It thus produces more mismatches than the NCC. This explains the worse correspondence matching rate of BICOS(+) when omitting the median filter (Figures 13 and 14). However, this problem is reduced by the filter.

### 3.3. Quality of the Coarse Correspondence Search

We calculated disparity maps for sequence lengths of 3 to 12, including the intermediate results before applying the $3 \times 3$ median filter for BICOS(+). We compared the coarse disparity maps to the ground truth disparity. Correspondences whose disparity differs more than 2 px from the ground truth were classified as "incorrectly matched". We show examples of which part of the test scene produces what percentage of missing or mismatched correspondences in Figure 13. In Figure 14 we show the combined percentages over the whole scene.

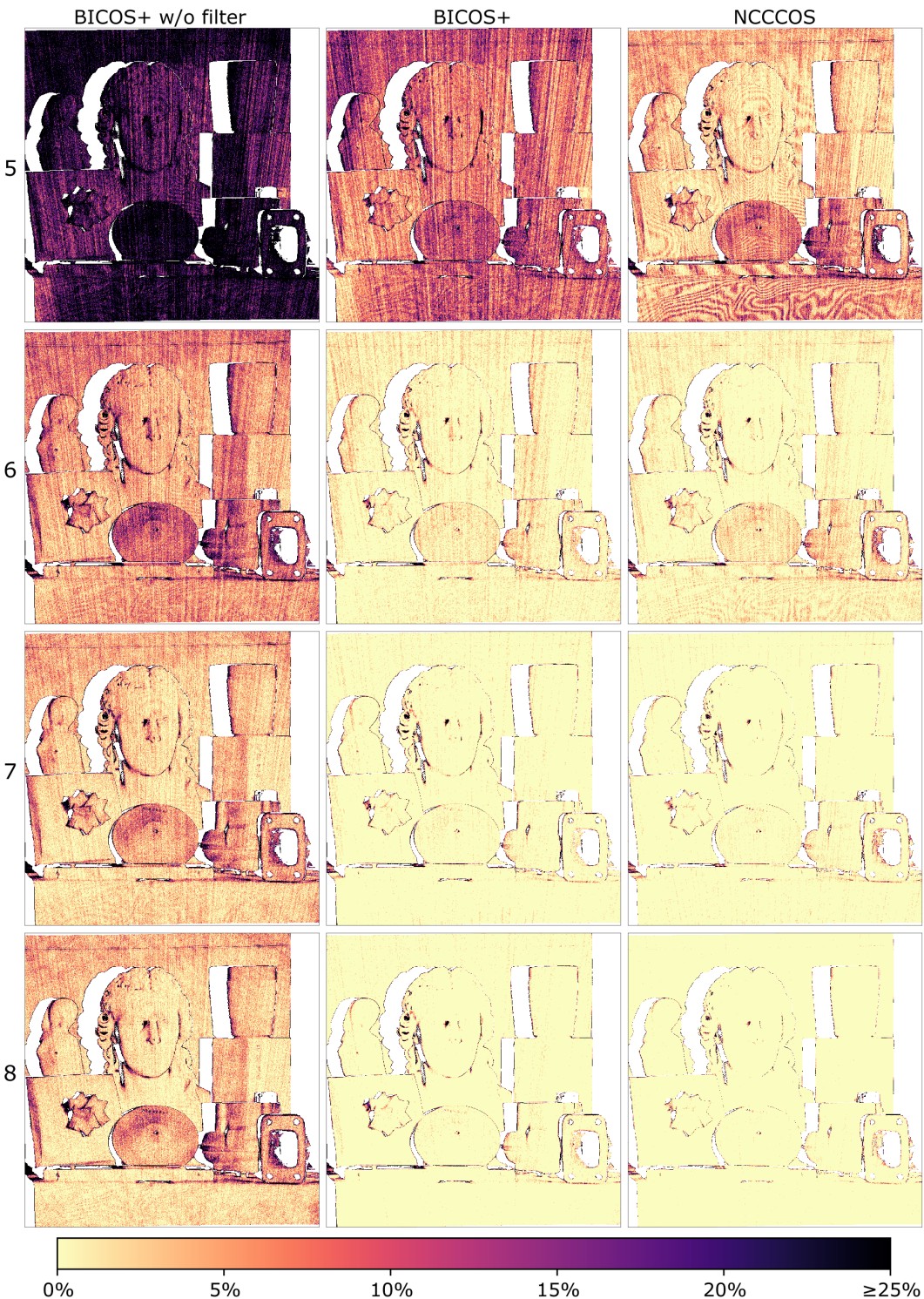

**Figure 13.** Percentage of wrong or missing correspondences (out of 39 data sets). Top to bottom: sequence lengths 5, 6, 7 and 8. Left to right: BICOS+ without median filter, BICOS+ (with median filter) and NCCCOS.

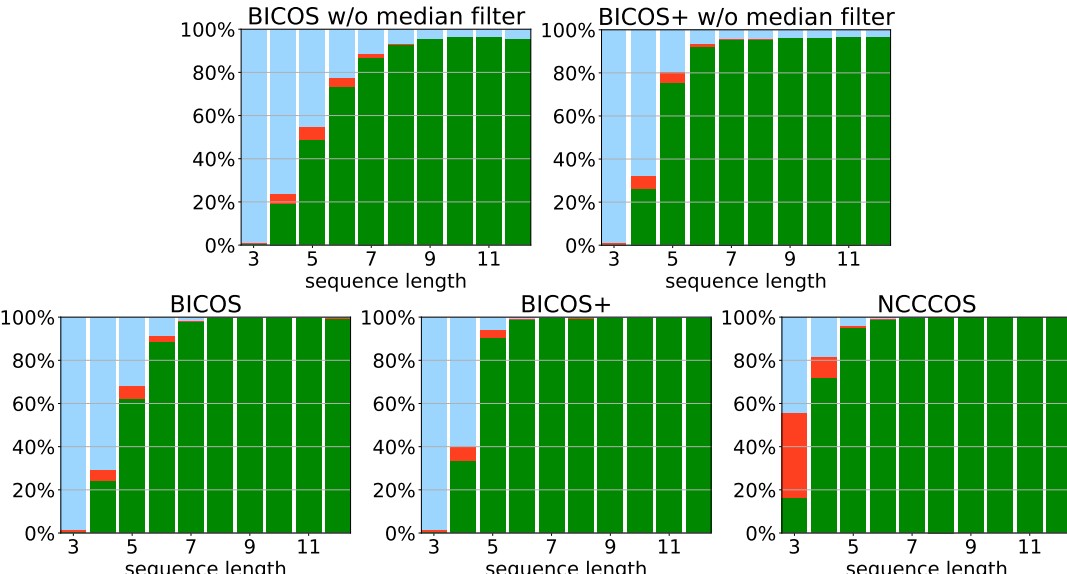

**Figure 14.** Quality of coarse correspondence search: green: percentage of correctly found correspondences, red: percentage of incorrectly found correspondences, blue: percentage of not found correspondences.

### 3.4. Speed of the Coarse Correspondence Search

We assessed the calculation speed of the algorithms. The BICOS and BICOS+ algorithms have the same speed because they only differ in the type of binary features which they use. We use OpenCL-based implementations of the algorithms and run them on NVIDIA GeForce 1080 graphics chips. We made 100 3D measurements of the test scene with a sequence length of 10 and recorded the calculation times of each part of the algorithms with the NVIDIA NSight profiler. Figure 15 shows the average calculation times. While the NCCCOS algorithm takes $(365 \pm 12)$ ms for rectification and coarse correspondence search, our BICOS(+) algorithm only takes $(19.0 \pm 1.8)$ ms (including rectification, pre-calculation of binary features and median filter).

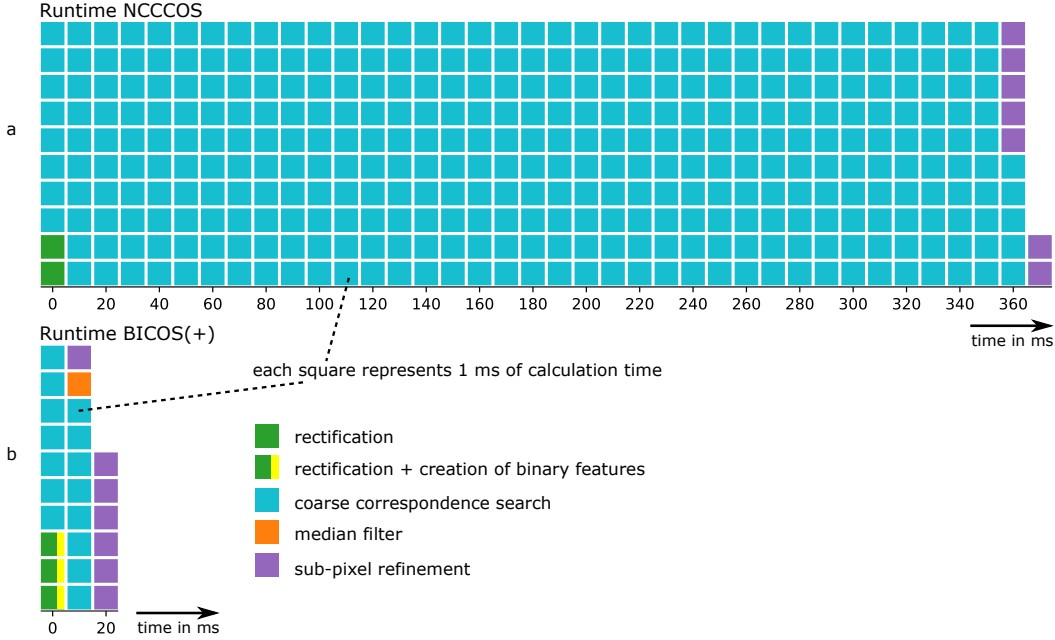

**Figure 15.** Algorithm runtime on NVIDIA GeForce 1080; each square represents one millisecond. (**a**) runtime of NCCCOS (reference algorithm), (**b**) runtime of BICOS(+).

## 4. Discussion

### 4.1. Interpretation of Results

As we show in Figure 11, when comparing two pixels, the pixel similarity measures NCC and NEBF are closely related to each other. However, the measure margins between correct and incorrect correspondences are better for the NCC than the NEBF (Figure 12). We think that this is a consequence of the reduction of information content in the binary features compared to the original brightness values. Therefore, it cannot be expected that an NEBF-based algorithm can produce better quality than the NCCCOS algorithm. This leads to more false matches and non-matches compared to the NCC-based correspondence search. The additional median filter step in BICOS(+) mostly compensates for this lower initial matching quality.

The effect of the signal to noise ratio (Figure 9c) on the coarse correspondence search result can be observed in Figure 13. The dark areas on the metal objects in the front of the scene have slightly increased error rates. A similar effect can be expected for a weaker projection light because it reduces the signal to noise ratio.

In comparison with the original BICOS algorithm, BICOS+ works on a higher number of binary features for sequence lengths of 10 and lower. This results in significant improvements at short sequence lengths of 7 and lower. The different way of calculating the binary features shows no disadvantages in the results.

Other ways of calculating binary features are possible, for example, comparing the sums of three or more brightness values. As the two types of binary features which we tested already show very similar results, we do not expect significant improvements with differently calculated binary features.

### 4.2. Comparison with Other Methods

Stereo vision with multi-shot statistical pattern projection is a comparatively small field of research. We know of no other publication on reconstruction algorithm speed for this special case. We hope to see more research in this field published in the future. In phase shift profilometry, correspondence search is not required, instead correspondence can be decoded from the brightness values of each single pixel. The computational workload is therefore significantly lower compared to our system with statistical patterns. For instance, Zhang and Huang [36] reported a computation time of 24.2 ms for the 3D-reconstruction of a scene with $532 \times 500$ pixels on a PC in the year 2004.

We did not asses the measurement accuracy of the final scan result in this contribution because the coarse correspondence search algorithms which we evaluated have little influence on it. The accuracy depends on the fine correspondence search which takes the coarse correspondence as an input. Schaffer et al. [15] have examined similar measurement accuracies for their statistical pattern system compared with a phase shift based system used by Zhang [37]. In previous publications, we have demonstrated the theoretical [17] and practical [18] equivalence of the statistical pattern projection method with the phase shift method in terms of accuracy.

## 5. Conclusions and Outlook

In this contribution, we described our algorithm Binary Correspondence Search (BICOS) and its improved variant BICOS+. It is a coarse correspondence search algorithm for multi-shot stereo sensors with statistical pattern projection. We compared it to the NCC-based reference algorithm (NCCCOS) in terms of speed and correctness of the coarse correspondence search result. Both algorithms share the same final correspondence refinement stage and therefore yield the same final results given a correct input from the coarse correspondence stage.

BICOS(+) is much faster than the NCCCOS algorithm. This results in a significantly reduced latency between image acquisition and availability of the 3D point cloud. While in an offline calculation scenario, this may be of little advantage, it is vital for applications like live patient monitoring.

We can thus conclude that for applications which require short latencies or high 3D frame rates, BICOS+ is the algorithm to use, while NCCCOS is to be preferred when 3D reconstruction speed is of no concern due to its slightly better coarse search correctness. If the additional flexibility which the statistical pattern method provides is not needed (i.e., much simpler projector construction, e.g., for extended spectral ranges or high-power, high-speed projection for large fields), classical phase shift profilometry can be a good alternative because it offers similar measurement accuracy and has much lower computational requirements.

In our future research, we want to investigate the influence of the projected patterns on the coarse correspondence search quality. Our goal is to achieve better results at shorter sequence lengths. This would allow us to reduce the total latency (of the image acquisition plus the reconstruction) even further.

**Supplementary Materials:** The rectified camera images which we used for our experiments are available online at https://drive.google.com/file/d/16E-vWnBAJGZZuj-labq4GPqjl7vSlq00/view?usp=sharing.

**Author Contributions:** Conceptualization, P.D., S.H., M.L.; methodology, P.D.; software, P.D.; validation, P.D.; formal analysis, P.D.; investigation, P.D.; resources, P.K., G.N.; data curation, P.D.; writing—original draft preparation, P.D.; writing—review and editing, P.D., S.H., M.L., P.K.; visualization, P.D.; supervision, G.N.; project administration, P.D.; funding acquisition, P.K., G.N.

**Funding:** This research received no external funding.

**Conflicts of Interest:** The authors declare no conflict of interest.

## Abbreviations

The following abbreviations are used in this manuscript:

Abbreviations

| | |
|---|---|
| BICOS | BInary Correspondence Search |
| BICOS+ | Improved Binary Correspondence Search |
| GOBO | GOes Before Optics (rotating slide) |
| NCC | Normalized Cross Correlation |
| NCCCOS | NCC-based Correspondence Search (the reference algorithm) |
| NEBF | Number of Equal Binary Features |
| NIR | Near-InfraRed light |
| SAD | Sum of Absolute Differences |

Mathematical Symbols

| | |
|---|---|
| $r$ | row in a rectified image |
| $c_L$ | column in a rectified image of the left camera |
| $c_R$ | column in a rectified image of the right camera |
| $d$ | the disparity between an object point in a left and in a right rectified image $d = c_L - c_R$ |
| $\vec{a} = a_1, ..., a_n$ $\vec{b} = b_1, ..., b_n$ | the brightness values of a pixel in an image sequence which consists of n rectified images |
| $A$ | ambient light |
| $s$ | factor representing the difference in camera sensitivity and reflectivity of the measured object towards the left and the right camera |

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
