# Peer review of "BICOS—An Algorithm for Fast Real-Time Correspondence Search for Statistical Pattern Projection-Based Active Stereo Sensors"

_applsci, doi:10.3390/app9163330_

Round 1

Reviewer 1 Report

 In this paper, the authors have reported a new BICOS(+) algorithm for the optical reconstruction of the object 3D geometric profile based on the pattern projection measurements system. The study is very interesting, however, several concerns in the current stage of manuscript need to be addressed before a consideration of acceptance, as follows:

1. Does the brightness of the projector influences the accuracy of the reconstruction? If yes, could you please add a brief analysis of the projector brightness effects?

2. If more cameras with different positions and angles are added, will the reconstruction better?

3. To make the manuscript self-contained, make sure all the variables have been fully specified. To make it more readable, I suggest to add a table list for the specification the variables?

4. Carefully recheck the grammar errors.

5. In the introduction part, the background of the works from other researchers should be added. The motivation or the merits of the contactless measurement should be added,

Relevant references and statements should be included. For example,

IEEE Trans. Instrum. Meas. 2016, 65, 2773; IEEE Trans. Instrum. Meas. 2017, 67, 167; NDT E Int 2018, 95, 36; Meas. 2017, 101, 118; Eur J Agron 2014, 55, 89; Biomed Opt Express 2010, 1, 471; Opt Nanoscopy 2012 1 6; Medical image analysis 2013 17 974; Sensors 2012 12 16785; Sensors 2011 11 90;

Author Response

Dear reviewer,

first of all, we would like to thank you for your comments on our paper "BICOS - an Algorithm for Fast Real Time Correspondence Search for Statistical Pattern Projection-based Active Stereo Sensors".

> 1. Does the brightness of the projector influences the accuracy of the reconstruction? If yes, could you please add a brief analysis of the projector brightness effects?

Thank you for the question. In order to answer it, we have recorded a different dataset with added inhomogeneous ambient light. We evaluated this dataset in terms of signal to noise ratio (Figure 10 c). In addition, we now show more intermediate algorithm results (Figure 13) and discuss them in the context of signal to noise ratio, including the expected effects of weaker projection light.

Please excuse that we did not reduce the projection light as this is technically challenging with our available setup, at the moment. However, the low reflectivity of the metal objects in the front of the scene has a very similar effect.

> 2. If more cameras with different positions and angles are added, will the reconstruction better?

The correspondence search algorithm exclusively works on a pair of cameras. Increasing the distance between the cameras would not influence the correspondence search algorithm directly. However, the accuracy of the final 3D model would increase (up to a view angle of 90 degrees between the cameras) because a correspondence error would then translate to a smaller 3D error in triangulation. In this case, the areas of the scene which are seen by both cameras together and illuminated by the projector get smaller. Therefore, the completeness of the scene would be reduced.

Additional cameras can be added e.g. to add color information to the 3D model. We have discussed this question in the author group and decided to not include the answer in the paper, because the paper focuses on the coarse correspondence search which is only a part of the full 3D reconstruction. The answer to this question would apply to the whole field of stereo vision, not only our algorithm.

> 3. To make the manuscript self-contained, make sure all the variables have been fully specified. To make it more readable, I suggest to add a table list for the specification the variables?

Thank you for the suggestion. This is a good idea. We have added a list of mathematical variables at the end of the paper after the list of abreviations.

> 4. Carefully recheck the grammar errors.

We have reworked a big part of the manuscript and also checked for grammar errors as good as we could.

> 5. In the introduction part, the background of the works from other researchers should be added. The motivation or the merits of the contactless measurement should be added, Relevant references and statements should be included.

Indeed, the introduction was until now very short and superficial. We have rewritten this part and now provide an overview over the context of our research. We also substantiate the introduction with many references.

Sections which we added or to which we applied major modifications:

1.1. Stereo vision based 3D sensors

1.2. Existing algorithms and algorithms in related fields

2.5. Robustness of the algorithms against ambient light and changes in reflectivity

2.7. Test scene and ground truth data

3.3. Quality of the coarse correspondence search

4.2. Comparison with other methods

5. Conclusions and outlook

Reviewer 2 Report

The paper describes an improved algorithm (a previous version of the same approach has been presented in a SPIE digital library paper this year) that reduce drastically the coarse identification of correspondences in an unstructured (a-priori unknown projection pattern) light system. The paper focus in particular on the performance comparison with slower algorithms (NCC-based), and on the evaluation of computational cost improvement of the new algorithm. Although not particularly original w.r.t. the original SPIE paper, the article is interesting, well written and well presented. Probably adding some more background references in the introduction section would improve the paper quality. At the same time, section 2.4 where the new algorithm (BICOS+) is presented could be better described. Even if the new algorithm simply consider a different set of comparisons to fill the binary features array, I think it would be beneficial to describe and comment more this section.

I found just a small typo at line 217 (“now disadvantages”).

Author Response

Dear reviewer,

first of all, we would like to thank you for your comments on our paper "BICOS - an Algorithm for Fast Real Time Correspondence Search for Statistical Pattern Projection-based Active Stereo Sensors".

> Probably adding some more background references in the introduction section would improve the paper quality.

Thank you for your motivation to extend and deepen the introduction. Indeed, the introduction was until now very short and superficial. We have rewritten this part and now provide an overview over the context of our research. We classify 3D sensors in three different ways such that the improvements due to our new algorithm can be assigned to the sensor classification. We also substantiate the introduction with many references.

> At the same time, section 2.4 where the new algorithm (BICOS+) is presented could be better described. Even if the new algorithm simply consider a different set of comparisons to fill the binary features array, I think it would be beneficial to describe and comment more this section.

Thank you also for this comment. We describe in detail the BICOS algorithm in the paragraph before. In the short paragraph about BICOS+ we only want to state the further development and emphasize the difference to the BICOS algorithm. Throughout the article we describe the background of both algorithms and we compare and discuss their benefits and drawbacks with respect to the NCCCOS reference algorithm.

We had a long discussion about your comment about BICOS+ within the author group. In the end, we came to the conclusion that if we enhanced this BICOS+ paragraph, we would only repeat information given in other parts of the article. However, we extended the result section with more intermediate results and added more discussion. We hope that this helps understanding the implications of the different algorithm variants.

Sections which we added or to which we applied major modifications:

1.1. Stereo vision based 3D sensors

1.2. Existing algorithms and algorithms in related fields

2.5. Robustness of the algorithms against ambient light and changes in reflectivity

2.7. Test scene and ground truth data

3.3. Quality of the coarse correspondence search

4.2. Comparison with other methods

5. Conclusions and outlook

Reviewer 3 Report

How about submitting this manuscript to MDPI Sensors journal?

The manuscript is extremely weak in terms of literature survey. Certains aspects of the proposed algorithm/topic area are incredibly popular; so it is tacit that a strong survey should be part of the paper. 

Of the 14 references in the bibliography, half of them are self-citations and the rest are basically textbooks. Is this paper so incredibly novel and isolated that there is nobody else working on even a related topic?!! Even if the core intellectual contribution is genuinely incredibly novel, there are several components of the entire computational pipeline that exist and have been extensively researched, therefore it is unacceptable to completely omit references for a scientific publication.

The authors need to motivate the research topic and their focus further. This is required for readers who are not already practitioners within the specific research niche of the manuscript.

The first thought that came to my mind half-way through the abstract was, "surely there are innumerable methods out there for correspondence search, so what is novel here; perhaps its a hardware acceleration based approach for low-powered devices".

It seems BICOS is already published in reference [5] as mentioned in line #36. This manuscript seems to be introducing BICOS+. So why not title the paper BICOS+ ? (Self-plagiarism warrants automatic rejection of manuscript!). The authors need to justify the novelty in the manuscript over and above their (or others) existing publications on this specific topic. It is based on the journal policy and editors, but about 30% novelty is recommended for incremental work.

Wide baseline stereo matching is one of the most intensely researched topics in computer vision and photogrammetry. It is essential for the authors to contrast and compare their algorithm against the top performing correspondence search methods in the field.

Empirical evaluation without a dataset isn't really possible. The authors should either find a good dataset for create a dataset for their empirical evaluation of their proposed algorithm.

The algorithm seems to be intensity based matching, which is not robust. There is a very large corpus of research in computer vision which deals with issues of intensity variation due to various reasons. The authors have not discussed how their algorithm is robust to variations or any kind. What is the payoff/applicability of this algorithm. Of course in designing data collection, these variations are included, so the algorithm can implicitly learn to model them and be robust to them for real world application.

Before the paper can be considered for acceptability, the authors must include in their manuscript:

(1) a proper literature survey

(2) at least one good dataset, preferably multiple datasets (includes noise and variation as might be expected in real world)

(3) comparative experimental evaluation

(4) Discuss the novelty and pay-off of their proposed algorithm in real-world application with its associated issues.

Author Response

Dear reviewer,

first of all, we would like to thank you for your comments on our paper "BICOS - an Algorithm for Fast Real Time Correspondence Search for Statistical Pattern Projection-based Active Stereo Sensors".

> The manuscript is extremely weak in terms of literature survey. Certains aspects of the proposed algorithm/topic area are incredibly popular; so it is tacit that a strong survey should be part of the paper.

> Of the 14 references in the bibliography, half of them are self-citations and the rest are basically textbooks. Is this paper so incredibly novel and isolated that there is nobody else working on even a related topic?!! Even if the core intellectual contribution is genuinely incredibly novel, there are several components of the entire computational pipeline that exist and have been extensively researched, therefore it is unacceptable to completely omit references for a scientific publication.

> The authors need to motivate the research topic and their focus further. This is required for readers who are not already practitioners within the specific research niche of the manuscript.

Thank you for your motivation to add more background information and literature survey. Indeed, the introduction was until now very short and superficial. We have completely rewritten this part and now provide an overview over the context of our research and related topics. We also substantiate the introduction with many references. In the discussion, we added a subsection "Comparison with other methods".

>The first thought that came to my mind half-way through the abstract was, "surely there are innumerable methods out there for correspondence search, so what is novel here; perhaps its a hardware acceleration based approach for low-powered devices".

> It seems BICOS is already published in reference [5] as mentioned in line #36. This manuscript seems to be introducing BICOS+. So why not title the paper BICOS+ ? (Self-plagiarism warrants automatic rejection of manuscript!). The authors need to justify the novelty in the manuscript over and above their (or others) existing publications on this specific topic. It is based on the journal policy and editors, but about 30% novelty is recommended for incremental work.

Please note that we do not claim a great novelty over our previously published conference paper.

However, MDPI Applied Sciences explicitly encourage the submission of research which was previously disclosed in conferences if it has not been peer reviewed (please refer to https://www.mdpi.com/journal/applsci/instructions section "Preprints and Conference Papers" for details).

We have added a note at the beginning of the manuscript to make it more explicit that we published a part of this research before.

> Wide baseline stereo matching is one of the most intensely researched topics in computer vision and photogrammetry. It is essential for the authors to contrast and compare their algorithm against the top performing correspondence search methods in the field.

We are sorry, that our previous version of this paper lacked the necessary background information.

We hope that the new sections 1.2. and 4.2. sufficiently inform the reader about this topic, now.

> Empirical evaluation without a dataset isn't really possible. The authors should either find a good dataset for create a dataset for their empirical evaluation of their proposed algorithm.

Thank you for the suggestion. As we do not know of any published dataset with statistical pattern projection, we have created a new dataset and publish it along with the paper.

The supplementary Data can be downloaded via the following link:

Since the IT security rules are very restrictive at our institute, this link will only be available for 48 hours. Sorry for the inconvenience.

> The algorithm seems to be intensity based matching, which is not robust. There is a very large corpus of research in computer vision which deals with issues of intensity variation due to various reasons. The authors have not discussed how their algorithm is robust to variations or any kind. What is the payoff/applicability of this algorithm. Of course in designing data collection, these variations are included, so the algorithm can implicitly learn to model them and be robust to them for real world application.

Thank you for the suggestion. We explain this matter in the new version of the script.

Intensity variations are not a significant problem, neither for our algorithm nor for the NCC based reference algorithm against which we compare ours. The main reason is that we work on temporal contrast, instead of spatial contrast, as in passive stereo vision.

We added Table 1 in this version, which sumarizes the main differences between the passive and our active method.

In the newly added section 2.5., we explain why the algorithms are robust against variations in reflectivity between the cameras and ambient light. To summarize it, the reference algorithm is robust, because the normalized cross correlation on which it is based normalizes offset and amplitude of the temporal brightness sequence. Our algorithm is robust because it works on binary features which are created by comparing two brightness values. Constant ambient light which is added to both values does not change the outcome of the comparison. And since the comparison yields a binary value, the scaling of the brightness values is invariant, too.

> Before the paper can be considered for acceptability, the authors must include in their manuscript:

> (1) a proper literature survey

> (2) at least one good dataset, preferably multiple datasets (includes noise and variation as might be expected in real world)

> (3) comparative experimental evaluation

> (4) Discuss the novelty and pay-off of their proposed algorithm in real-world application with its associated issues.

We hope that we have satisfied these requirements by adding/exchanging the following sections in the manuscript:

1.1. Stereo vision based 3D sensors

1.2. Existing algorithms and algorithms in related fields

2.5. Robustness of the algorithms against ambient light and changes in reflectivity

2.7. Test scene and ground truth data

3.3. Quality of the coarse correspondence search

4.2. Comparison with other methods

5. Conclusions and outlook

Reviewer 4 Report

The research studied the 3D reconstruction using a new method named as BICOS+. The proposed method is interesting, the results though lack of are interesting.  The following comments are provided:

(1) The introduction section should propose more rationals for why the research is needed. Appear to me that the authors just want to develop an extended method based on their previous paper. The question is why it is needed so much?

(2) The method is described in some detail, but without comparison back to other methods, it can be challenging for others to understand why the method is so interesting and where the innovation/breakthrough is.

(3) Results are lack of. Authors should present the 3D reconstruction results from NCCCOS method. So it is easier for the audience to justify the advantages of the proposed method.  It also only has one scene 3D reconstruction. More scenes 3D reconstruction should help to reinforce the applications of the proposed method. 

(4) From Figure 10, it appears that BICOS method has also its own problem compared to NCCCOS method. How the problem can be addressed?

(5) The discussion section should also discuss the comparison of the run-time, 3D reconstruction results with other methods. 

Author Response

Dear reviewer,

first of all, we would like to thank you for your comments on our paper ``BICOS - an Algorithm for Fast Real Time Correspondence Search for Statistical Pattern Projection-based Active Stereo Sensors''.

> (1) The introduction section should propose more rationals for why the research is needed. Appear to me that the authors just want to develop an extended method based on their previous paper. The question is why it is needed so much?

Thank you for your motivation to extend and deepen the introduction. Indeed, the introduction was until now very short and superficial. We have rewritten this part and now provide an overview over the context of our research. We also substantiate the introduction with many references.

Please note that we do not claim a great novelty over our previously published conference paper. However, MDPI Applied Sciences explicitly encourage the submission of research which was previously disclosed in conferences if it has not been peer reviewed (please refer to https://www.mdpi.com/journal/applsci/instructions section "Preprints and Conference Papers" for details). We have added a note at the beginning of the manuscript to make it more explicit that we published a part of this research before.

> (2) The method is described in some detail, but without comparison back to other methods, it can be challenging for others to understand why the method is so interesting and where the innovation/breakthrough is.

In addition to the introduction, we have added a subsection "Comparison with other methods" to the Discussion section. We also extended the Conclusions section.

> (3) Results are lack of. Authors should present the 3D reconstruction results from NCCCOS method. So it is easier for the audience to justify the advantages of the proposed method.  It also only has one scene 3D reconstruction. More scenes 3D reconstruction should help to reinforce the applications of the proposed method.

We agree that it was up to now difficult to see the differences between the algorithms in the results. We have discussed the additional display of refined 3D models in the author group and came to the conclusion that this would not add information about the difference in the coarse correspondence search, which is the focus of our paper. Instead, we now show where the errors in the coarse correspondence search happen in the added Figure 13. We hope this demonstrates the difference between the algorithms well.

> (4) From Figure 10, it appears that BICOS method has also its own problem compared to NCCCOS method. How the problem can be addressed?

Yes, you are absolutely right. We address this problem with the median filter at the end of the coarse search algorithm. We have changed the paper in several places to distinguish better between the intermediate result of the BICOS(+) algorithm and its final result after the median filter. In the result section, we now also show the intermediate results.

> (5) The discussion section should also discuss the comparison of the run-time, 3D reconstruction results with other methods.

We added this to the "Comparison with other methods" section in the Discussion section.

Sections which we added or to which we applied major modifications:

1.1. Stereo vision based 3D sensors

1.2. Existing algorithms and algorithms in related fields

2.5. Robustness of the algorithms against ambient light and changes in reflectivity

2.7. Test scene and ground truth data

3.3. Quality of the coarse correspondence search

4.2. Comparison with other methods

5. Conclusions and outlook

Round 2

Reviewer 3 Report

19: I appreciate the upfront disclaimer, but please move that line elsewhere in the introduction. It is unusual to have it as the first line in a paper.

Thank you for addressing some of the concerns raised in the previous review of the manuscript.

If MDPI does not insist on a certain magnitude of new original contributions over and above previous conference publication, I have no issues with that aspect. If the authors would like to, they can change the title from BICOS+ to BICOS.

Author Response

Dear reviewer,

Thank you for your review.

> I appreciate the upfront disclaimer, but please move that line elsewhere in the introduction. It is unusual to have it as the first line in a paper.

We agree with you on this point.

However, MDPI require the disclaimer to be on the first page of the manuscript.

Unfortunately, we didn't find a better place to include it because there are only a few available lines after the abstract.

> Thank you for addressing some of the concerns raised in the previous review of the manuscript.

> If MDPI does not insist on a certain magnitude of new original contributions over and above previous conference publication, I have no issues with that aspect. If the authors would like to, they can change the title from BICOS+ to BICOS.

We have changed the title back to "BICOS - .....".

Reviewer 4 Report

Authors have addressed my previous concerns thus I recommend for publication. 

Author Response

Dear reviewer,

we appreciate your recommendation.
Thank you.